# Adsorption Mechanism of Chloropropanol by Crystalline Nanocellulose

**DOI:** 10.3390/polym14091746

**Published:** 2022-04-25

**Authors:** Jinwei Zhao, Zhiqiang Gong, Can Chen, Chen Liang, Lin Huang, Meijiao Huang, Chengrong Qin, Shuangfei Wang

**Affiliations:** Guangxi Key Laboratory of Clean Pulp & Papermaking and Pollution Control, School of Light Industrial and Food Engineering, Guangxi University, Nanning 530004, China; 1916391038@st.gxu.edu.cn (J.Z.); 15038502915@163.com (Z.G.); cc15125007558@163.com (C.C.); huanglin2841@163.com (L.H.); hmj0026@163.com (M.H.); qin_chengrong@163.com (C.Q.); wangsf@gxu.edu.cn (S.W.)

**Keywords:** paper packaging, chloropropanol, crystalline nanocellulose, adsorption kinetics, quartz crystal microbalance

## Abstract

Paper packaging materials are widely used as sustainable green materials in food packaging. The production or processing of paper materials is conducted in an environment that contains organic chlorides; therefore, potential food safety issues exist. In this study, the adsorption behavior of organic chlorides on paper materials was investigated. Chloropropanol, which has been extensively studied in the field of food safety, was employed as the research object. We studied the adsorption mechanism of chloropropanol on a crystalline nanocellulose (CNC) model. The results demonstrated that physical adsorption was the prevailing process, and the intermolecular hydrogen bonds acted as the driving force for adsorption. The adsorption effect assumed greatest significance under neutral and weakly alkaline conditions. A good linear relationship between the amount of chloropropanol adsorbed and the amount of CNC used was discovered. Thus, the findings of this study are crucial in monitoring the safety of products in systems containing chloropropanol and other chlorinated organic substances. This is particularly critical in the production of food-grade paper packaging materials.

## 1. Introduction

Plastic packaging offers protection and has a low packaging weight. Both factors have a positive impact on transportation and a longer service life. Therefore, plastic materials are widely used in food packaging [1]. Between 2018 and 2019, global plastic production reached 359 million tons [2], of which approximately 40% was used for packaging [3]. However, the applicability of plastic materials has been increasingly restricted owing to the non-degradability of plastics, release of hazardous chemicals during the process of manufacturing and use, as well as environmental pollution caused by landfills, incineration, or improper treatment after disposal [4,5,6]. Paper packaging materials possess the advantages of renewable resources [7] such as easy waste recycling ability and degradability [8]. They are considered the most promising green packaging materials. This can potentially play an important role both in the present and in the future [9]. The replacement of plastic products with paper packaging follows a recent development trend. Alternative products, such as paper bags, paper tableware, and paper straws, have been developed.

Food packaging materials provide physical protection to food. They exhibit excellent barrier properties as well as mechanical and optical properties. This helps in ensuring sufficient shelf life while maintaining the safety and quality of the packaged food [10]. However, chemical additives are added in the production of paper food packaging materials. Additionally, hazardous substances are present in the production environment [11,12]. These substances are at a risk of migrating into the packaged food. Therefore, the migration of hazardous materials in paper packaging is a major issue in the area of food safety. Among these substances, chloropropanols are the most common. Chloropropanols are toxic to various organisms [13]. They can attack various organs, cause diseases of various tissues, and affect the normal functions of living organisms. Therefore, several countries worldwide have conducted extensive research on the content and migration of chloropropanols in food and have imposed strict restrictions on them. Both the U.S. Food and Drug Administration and European Commission have stipulated limits on the content of chloropropanol in foods and food additives.

With the widespread applications of paper food packaging, the use of chloropropanol in paper materials has also received considerable attention [14]. Existing research has revealed that there exist two primary sources of organic chlorides in paper materials. One source is organic chlorides, introduced by various additives added in the production of paper food packaging materials. For example, polyamideamine epichlorohydrin (PAE) is often added to paper food packaging materials to improve their moisture resistance. This type of polymer with epichlorohydrin as a raw material also produces 3-monochloropropane diol and 1,3-dichloro-2-propanol during the process [15]. The other source is chlorine-containing organic matter in the production system, which enters the material via adsorption and other methods. Recent studies have discovered that paper materials produce chloropropanols during the chlorine-containing bleaching process [16]. White cardboard plays an important role in paper food packaging materials. White cardboard is a paper product made from bleached chemical pulp and a high-yield pulp. It possesses a high stiffness and surface strength and is widely used in food packaging materials [17,18]. Bleached chemical pulp is rich in sulfonic acid groups, whereas the high-yield pulp contains more carboxyl groups. The former is rich in sulfonic acid groups, whereas the latter contains more carboxyl groups. These two groups exhibit strong adsorptivity. Nanocellulose with sulfonic acid groups is often used for the adsorption of heavy metal ions in sewage and treatment of polluted water containing amines [19]. It exhibits a good adsorption effect. Carboxylated nanocellulose possesses the characteristics of a large specific surface area and high functional group density [20]. It is often used in environmental remediation and strongly adsorbs dye molecules and heavy-metal ions [21]. Therefore, it is useful in the production of paper-packaging materials. It can potentially adsorb the by-products of chemical additives and chlorinated organics in bleaching wastewater, which may cause potential food safety risks.

The adsorption process and properties of the adsorption layer are usually interpreted at the molecular level using methods such as quartz crystal microbalance (QCM) or ellipsometry [22]. QCM has ng-level quality inspection capabilities. In recent years, it has been widely used in the study of interfacial adsorption behavior [23]. The adsorption behavior of a composite material with crystalline nanocellulose (CNC) as the substrate for different molecules and ions was studied using QCM-D [24,25]. QCM-D was successfully applied to the adsorption of PAE on the surface of cellulose to improve the wet and dry strength of paper and effectively explain the adsorption behavior of amphoteric electrolytes. The adsorption performance of sulfonated nanocellulose with regard to metal ions was considerably enhanced by increasing the degree of substitution and by the dynamic process of adsorption and analysis of the same cellulase on cellulose.

Two types of CNC were selected for this study. Chloropropanol was used as a typical simulant of chloride organics. The possible adsorption mechanism and behavior of cellulose fibers on chloropropanol were investigated. We analyzed the samples using QCM-D, an atomic force microscope (AFM), a scanning electron microscope (SEM), and an electrophoretic light scattering analyzer (DELSA). The solution pH and CNC volume were considered in this study. Factors such as the influence of chloropropanol concentration on the adsorption capacity of CNC were investigated. Therefore, this provides a reference for the application of paper packaging materials in food packaging materials. To explore the adsorption capacity and adsorption mechanism of chlorine-containing organic compounds in paper materials, and to provide a theoretical basis for reducing the adsorption capacity of chlorine-containing organic compounds in paper materials.

## 2. Materials and Methods

### 2.1. Materials

Sulfonated nanocellulose (diameter 4–10 nm, length 100–500 nm) and carboxylated nanocellulose (diameter 4–10 nm, length 100–500 nm) were purchased from Macleans Reagent Co., Ltd. (Shanghai, China). Polyethylenimine (PEI, *M*_w_ = 7.5 × 10^5^, 50 wt% in H_2_O) was purchased from Aladdin Reagent Company (Shanghai, China). Chloropropanol (98%) was purchased from Sigma reagent (Shanghai, China).

### 2.2. Sample Configuration

PEI was dissolved in Milli-Q and stirred for 30 min to prepare a 0.2% PEI solution. Three uniformly dispersed CNC suspensions with concentrations of 0.01%, 0.05%, and 0.1%, respectively, were prepared with deionized water, which was dispersed at 3000 rpm for 5 min, using a T25 digital package high-speed disperser (T25, IKA, Guangzhou, China). Chloropropanol solutions of 0.01%, 0.02%, 0.05%, and 0.1% were prepared. All liquid solutions were bubbled using N_2_ gas. Subsequently, they were degassed for 30 min and stored in a refrigerator at 4 °C.

### 2.3. QCM Detection

SiO_2_-coated chips were cleaned with 0.2% sodium dodecyl sulfate (>60 min), rinsed with Milli-Q water, dried in the presence of N2, and treated with ultraviolet ozone (15 min). A crystal sensor (Biolin Scientific, Gothenburg, Sweden) was placed in the QCM (QHM401,Biolin Scientific, Gothenburg, Sweden) instrument, and deionized water was introduced into the QCM channel at a rate of 100 μL·min^−1^ to run the baseline. After the baseline was stable, the PEI solution was injected into the QCM channel. When the signal was stable, the CNC solution was injected into the channel. Subsequently, chloropropanol was injected into the QCM channel until the signal was stable, and the adsorption process was monitored. The frequency and dissipation of the third, fifth, and seventh channels were recorded to further analyze the adsorption kinetics of chloropropanol on CNC. The adsorbed quantity was usually very low, resulting in small differences between the channels. The calculations in this study were analyzed using the third channel frequency and Sauerbrey’s equation. The QCM adsorption experiment was significantly impacted at a temperature of 25 °C owing to the density of the fluid in the QCM instrument [26]. Therefore, the temperature was maintained at 25 ± 0.02 °C in this study. The flow rate was maintained at 0.1 mL·min^−1^. All tests were conducted in parallel for 3 times. The error range of CNC adsorption amount is between 1.4% and 4.5%. The error range of the adsorption capacity of chloropropanol was between 2.8% and 12.5%.

### 2.4. SEM Characterization

After the experiment, the SiO_2_ sensor ((Biolin Scientific, Gothenburg, Sweden)was vacuum dried for 24 h. It was sprayed with gold using sputtering coating machine (Vapor Technologies, Longmont, CO, USA). The microstructure and size of CNC adsorbed onto the SiO_2_ sensor were analyzed using SEM (TESCAN MIRA4, TESCAN, Brno, Czech Republic). The SEM was equipped with an energy spectrometer (EDS) (TESCAN, Brno, Czech Republic) to map the elements of the sample and analyze the relevant components.

### 2.5. AFM Characterization

CNC was analyzed before and after adsorption on the SiO_2_ sensor using an AFM (Dimension Edge, Bruker Co, Ltd., Berlin, Germany). Commercial probes were used with a spring constant of 20–80 N·m^−1^ and a resonant frequency of 300–340 kHz. The scanned dimensions were 1 × 1 µm^2^ and 1 × 1 nm^2^.

### 2.6. Zeta Potential Measurement

The zeta potentials of the CNC suspensions and chloropropanol solutions at different pH values were studied using an electrophoretic light scattering analyzer (DelsaMax, Beckman Coulter, Indianapolis, IN, USA). The stability of the suspension and the functional groups on the surface of the nanocellulose were analyzed. A homogeneous suspension (25 mL) was tested at 20 °C. The accuracy of this method was 5% [27].

## 3. Results and Discussion

### 3.1. Surface Analysis before and after CNC Adsorption

The adsorption of CNC on the SiO_2_ sensor was analyzed using an SEM. The results are demonstrated in Figure 1a. CNC forms a uniformly distributed thin film on the SiO_2_ sensor. The cellulose nanocrystals are tightly interwoven. A large number of micropores are distributed on the nanocellulose coating, with a diameter of approximately 0.05 μm, as depicted in Figure 1b,c. This can be attributed to the large number of hydroxyl groups in CNC. The high density of hydroxyl groups allows CNC to be combined through hydrogen bonds. A strong porous structure is formed. Following the adsorption of CNC, the SiO_2_ sensor attains a certain capacity of physical adsorption. As pointed out by Shuai Li in 2011, porous films have physical adsorption capacity [28].

The dimensional information and surface morphology were observed using an AFM. From the AFM image of the stable adsorption of CNC on the SiO_2_ sensor (Figure 2a), it can be observed that the surface of the CNC film is not uniform and smooth. There exist voids of approximately 50 nm; this is consistent with the SEM data. Figure 2b illustrates the accumulation of CNC on the surface of the SiO_2_ sensor. We can observe that the surface of the CNC film is irregular. Combined with the longitudinal section data analysis, it can be observed that there exists a certain height difference on the surface of the cellulose film (Figure 2d). Additionally, there exists a height difference between the fiber bundles of the clusters (Figure 2e) [29].

The CNC film on the SiO_2_ sensor was analyzed before and after the adsorption of chloropropanol. The results of the EDS energy spectrum analysis are illustrated in Figure 3. Figure 3b reveals that the energy spectrum of the CNC film, after the addition of the chloropropanol solution, is densely packed with chlorine. (The red dots represent chlorine). However, the energy spectrum of the CNC film illustrated in Figure 3e, before the addition of the chloropropanol solution, depicts only scattered chlorine elements. The mapping chlorine analysis image (Figure 3b) and the original SEM image (Figure 3a) were overlapped to obtain Figure 3c. Figure 3d,e were combined to obtain Figure 3f. Next, Figure 3c,f were compared and analyzed. As can be observed, the chloropropanol solution is adsorbed after it is passed through the CNC film. No chlorine distribution can be observed in the CNC accumulation holes. Chlorine is intensively adsorbed onto the fiber. Therefore, chloropropanol was stably adsorbed on the surface of the CNC. Moreover, the adsorption capacity was high.

### 3.2. Effect of Chloropropanol Concentration on the Adsorption Capacity of CNC

We investigated the extent to which the adsorption effect on CNC was influenced by the concentration of chloropropanol. The adsorption capacity of CNC was maintained at a constant value by controlling the concentration of the CNC solution in the QCM. The results are depicted in Figure 4. At CNC concentrations of 0.01%, 0.02%, 0.05%, and 0.1%, the SiO_2_ sensor adsorbs 740 ± 20 ng, 1240 ± 40 ng, 1780 ± 80 ng, and 2130 ± 30 ng of CNC, respectively. Different concentrations of chloropropanol were passed into CNC films of the same quality. The amount of chloropropanol adsorbed on the CNC film was maintained at 120 ± 15 ng, 330 ± 30 ng, 350 ± 10 ng, and 680 ± 30 ng, respectively. The change in the concentration of chloropropanol had a minimal effect on the chloropropanol adsorption capacity of CNC when the amount of CNC was constant. The same amount of CNC adsorbed the same amount of chloropropene at different concentrations of the chloropropanol solution. Therefore, the concentration of chloropropanol did not significantly affect its adsorption on the nanofibers. The concentration of chloropropanol is, therefore, not a major influencing factor in the process of adsorption on CNC.

### 3.3. Effect of CNC Dosage pH on the Adsorption Capacity

The dosage of CNC on the SiO_2_ sensor was controlled by adjusting the concentration of the CNC solution in the QCM. Figure 4b indicates that as the adsorption capacity of CNC increases, the adsorption capacity of CNC for chloropropanol simultaneously increases. The relationship between the adsorption amount of CNC and the amount of chloropropanol adsorbed was studied. The results in Figure 5b indicate that the adsorption of chloropropanol on sulfonated CNC exhibits a good linear relationship under different pH conditions. Thus, it can be concluded that there exists a good linear relationship between the adsorption capacities of CNC and chloropropanol. To eliminate the influence of sulfonic acid groups, carboxylated nanocellulose was selected. The obtained results under the same experimental conditions are presented in Figure 5. From Figure 5a,b, we can observe that the quantity of chloropropanol does not start from zero. This can be attributed to the presence of hydrophilic groups (–OH) on the CNC surface. The water molecules in the solution are associated with the hydroxyl groups in the form of hydrogen bonds. A hydration layer with a thickness equivalent to that of water molecules is formed on the membrane surface. The water molecules in the hydration layer are highly structured. They are in a dynamic equilibrium with free water molecules. Therefore, chloropropanol is not effectively adsorbed at the beginning of the CNC adsorption process. The initial mass is the mass of the hydration layer. In summary, the adsorption capacity of CNC is an important factor in determining the adsorption capacity of chloropropanol. There exists a good linear relationship between the adsorption capacities of CNC and chloropropanol. The adsorption capacity of chloropropanol increases with an increase in the CNC adsorption capacity.

### 3.4. Effect of pH on the Adsorption Capacity

As presented in Figure 5a,b, the adsorbed amount of chloropropanol, with the same CNC adsorption capacity, can be changed by adjusting the pH. Under varying pH conditions, the linear relationship between the adsorption amount of CNC and the adsorption amount of chloropropanol was found to be different. Therefore, we studied the influence of the pH conditions on the adsorption of chloropropanol on CNC. Figure 6 depicts the amount of chloropropanol adsorbed per unit volume of CNC under different pH conditions for the two types of CNC. The pH of the solution significantly affects the adsorption of chloropropanol on CNC; the adsorption is optimized in neutral and weakly alkaline environments. An acidic environment inhibits the adsorption of chloropropanol on CNC. When the pH of the adsorption system is approximately four, the chloropropanol adsorption capacity of CNC is zero. With an increase in the pH value, the adsorption capacity of CNC is gradually increased. However, when the alkalinity is strong, the adsorption of chloropropanol on CNC is inhibited. This is because the adsorption process of CNC on chloropropanol involves physical adsorption. No charge was observed on the surface of the chloropropanol. Therefore, the driving force between the CNC and chloropropanol molecules can be attributed to the intermolecular hydrogen bond and van der Waals forces. Particularly, the number of donors and acceptors of the hydrogen bond in chloropropanol are both one. The adsorption of chloropropanol on CNC is inhibited under acidic conditions owing to the existence of sulfonic acid groups (R–SO_3_^−^) and carboxyl groups (R–COO^−^), which alter the ionization balance equation. The number of free hydrogen ions is increased, negative atoms on both sides are covered, and formation of hydrogen bonds is inhibited. Under strongly alkaline conditions, there exist excess hydroxide radicals in the solution, and the hydrolysis equilibrium shifts to the right owing to the active free hydrogen ions being covered. Hydrogen bond formation is inhibited by the absence of sufficient free hydrogen ions. This results in the inhibition of chloropropanol adsorption and a reduction in the adsorption capacity of CNC [30].

### 3.5. Adsorption Kinetics

The adsorption behavior of chloropropanol on the CNC model was investigated. Figure 7a illustrates the frequency of chloropropanol adsorption on CNC as measured by QCM. The adsorption of chloropropanol proceeds rapidly, and the entire adsorption process occurs in 200 s. This is because there are several adsorption sites on the CNC surface; the adsorption driving force is large, and there is no competition between the adsorption molecules. Therefore, chloropropanol can be adsorbed quickly on the CNC model. The adsorption gradually approaches an equilibrium after 200 s. This is because the adsorption driving force between the two phases was weakened, the adsorption active sites were occupied by chloropropanol molecules, and the adsorption process gradually reached the equilibrium state. Therefore, the adsorption capacity for chloropropanol remains unchanged after 200 s.

Usually, the study of the adsorption kinetic process simulation of a solid-liquid two-phase system is composed of the intra-particle diffusion model equation, quasi-first-order kinetic equation, and quasi-second-order kinetic equation. The quasi-first-order kinetic model can be expressed as follows [31]:(1)$$\ln\left(q_{e}-q_{t}\right)=\ln q_{e}-k_{1t}$$
where *k*_1_ is the rate constant of quasi-first-order adsorption. *q_t_* and *q_e_* are the amounts of chloropropanol adsorbed on the CNC model at time *t* and at equilibrium, respectively.

The quasi-second-order kinetic model can be represented by the following equation:(2)$$t/q_{t}=1/k_{2}q_{e}^{2}+t/q_{e}\;$$
where *k*_2_ is the rate constant of quasi-second order adsorption.

The intraparticle diffusion model can be defined by the following equation
(3)$$q_{t}=k_{i}t_{1}/2+C\;$$
where *k_i_* is the rate constant of the intraparticle diffusion model, and *C* is a constant related to the thickness of the boundary layer.

These three equations were applied to simulate the CNC adsorption process of chloropropanol molecules. When the particle diffusion simulation equation was used, linear fitting of the equation was not ideal. Therefore, the quasi-first-order kinetic model was used for the simulation; consequently, it demonstrated a better fit. The fitted image is depicted in Figure 7b. The fitting was found to be poor when the quasi-second-order kinetic equation was adopted to analyze the adsorption data. Therefore, it can be concluded that the adsorption of chloropropanol on CNC ideally follows a quasi-first-order adsorption curve.

The driving force for the adsorption of chloropropanol was studied, and zeta potentials of CNC solutions under different pH conditions were determined. For CNC solutions containing carboxyl groups, the zeta potential moved in a positive direction with an increase in pH. For CNC solutions containing sulfonic acid groups, the zeta potentials did not significantly vary at different pH conditions. This is inconsistent with the changing trend observed in the ratio of the adsorption capacity of chloropropanol to that of CNC. This indicates that the zeta potential of the CNC solution has no significant effect on the adsorption of chloropropanol. Therefore, it can be inferred that the driving force of adsorption between CNC and chloropropanol is not electrostatic but an intermolecular force. In summary, the adsorption of chloropropanol on CNC relies primarily on intermolecular forces. The entire adsorption process is primarily physical, and the adsorption process can be described by a quasi-first-order kinetic equation.

### 3.6. Saturated Adsorption Capacity of Chloropropanol by CNC

The maximum adsorbed amount of chloropropanol per unit mass of CNC was studied. Table 1 lists the amount of chloropropanol adsorbed per unit mass of the two types of CNC under different pH conditions, which were studied using quartz microbalance technology. As presented in Table 1, the adsorption effect of chloropropanol per unit mass of CNC is superior. This is considerably higher than the limit of adsorbable organic halogen content of paper in China’s national standard GB/T36420–2018 (15 mg·kg^−1^ or 1.5 × 10^−5^), especially under weakly alkaline and neutral conditions; it is 20,000 times that of the allowable limit. Therefore, it is important to control the concentration of organic chlorides in paper production systems. Modern papermaking processes are typically performed under weakly alkaline conditions [18]. Chlorinated organics are extremely likely to be encountered during paper production. The fiber adsorption speed can achieve adsorption saturation in a short time. This is extremely likely to cause excessive levels of chloropropanol in paper and food packaging. Therefore, the potential risks are worthy of attention.

## 4. Conclusions

The adsorption behavior of chloropropanol on CNC was studied, and the adsorption kinetics was established. The morphology of CNC and distribution of chloropropanol were analyzed after the introduction of chloropropanol. The results indicate that chloropropanol is physically adsorbed on CNC. Moreover, intermolecular hydrogen bonds are the primary driving force of adsorption. The adsorption process conforms to the quasi-first-order kinetic equation. The adsorption capacity of CNC and the pH of the solution are critical factors that affect the adsorption of chloropropanol. There exists a good linear relationship between the adsorption capacity of CNC and adsorption capacity of chloropropanol. The amount of chloropropanol adsorbed correspondingly increases with the amount of CNC used. The adsorption of chloropropanol on CNC is optimal under weak alkali and neutral conditions. The maximum adsorption capacity is 20,000-times that of the national standard; therefore, the potential risk caused is noteworthy.

## Figures and Tables

**Figure 1 polymers-14-01746-f001:**
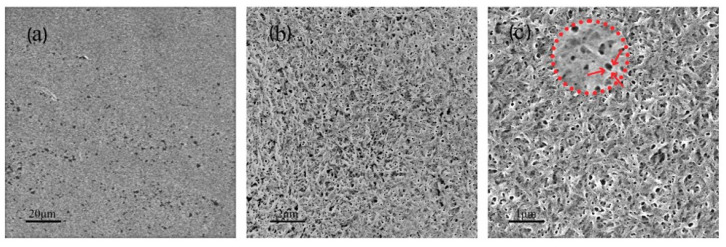
SEM of SiO_2_ sensor before and after CNC adsorption.

**Figure 2 polymers-14-01746-f002:**
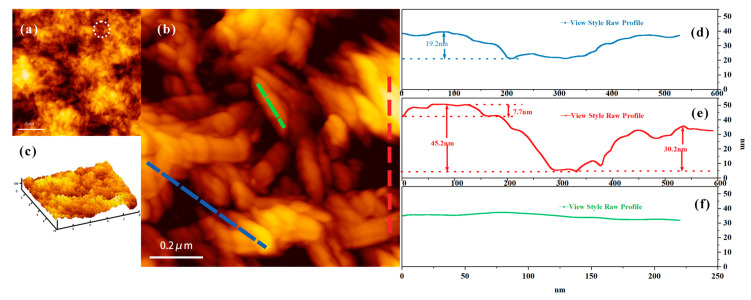
AFM morphology of CNC and the cross section of the fiber. ((**c**) is 3D image of CNC film. (**d**–**f**) are the length and section height data of the corresponding color line segment in C respectively. The three line segments of red, yellow and blue are randomly selected, and the height difference of the surface of nano cellulose film is measured).

**Figure 3 polymers-14-01746-f003:**
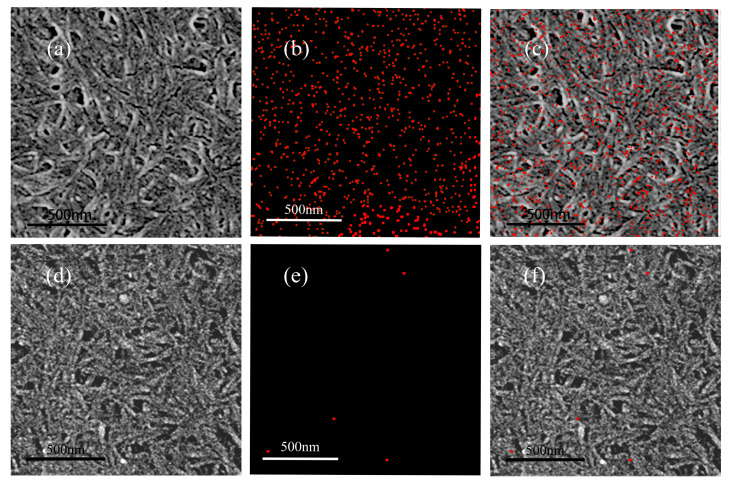
Mapping analysis of CNC before and after the adsorption of chloropropanol.

**Figure 4 polymers-14-01746-f004:**
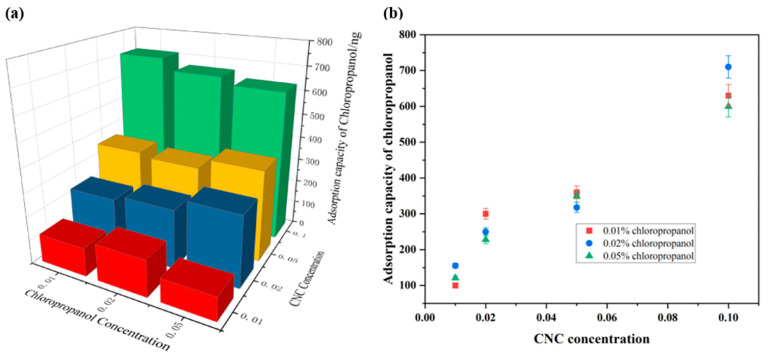
The effect of chloropropanol concentration on its adsorption on CNC. (**a**) The effects of CNC concentration and chloropropanol concentration on the adsorption capacity of chloropropanol; (**b**) The effect of CNC Concentration on Adsorption Capacity of Chloropropanol.

**Figure 5 polymers-14-01746-f005:**
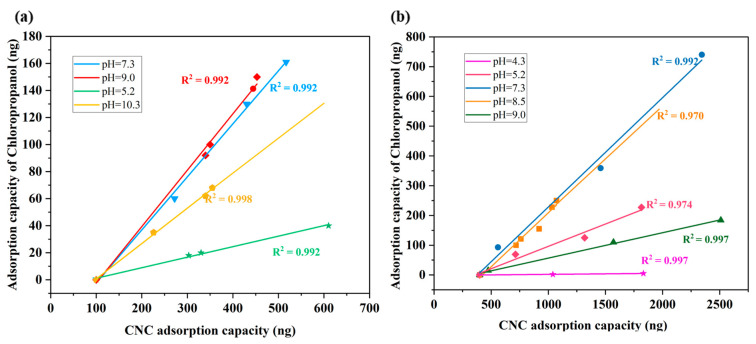
Adsorption capacity of chloropropanol on sulfonated and carboxylated nanocellulose samples.

**Figure 6 polymers-14-01746-f006:**
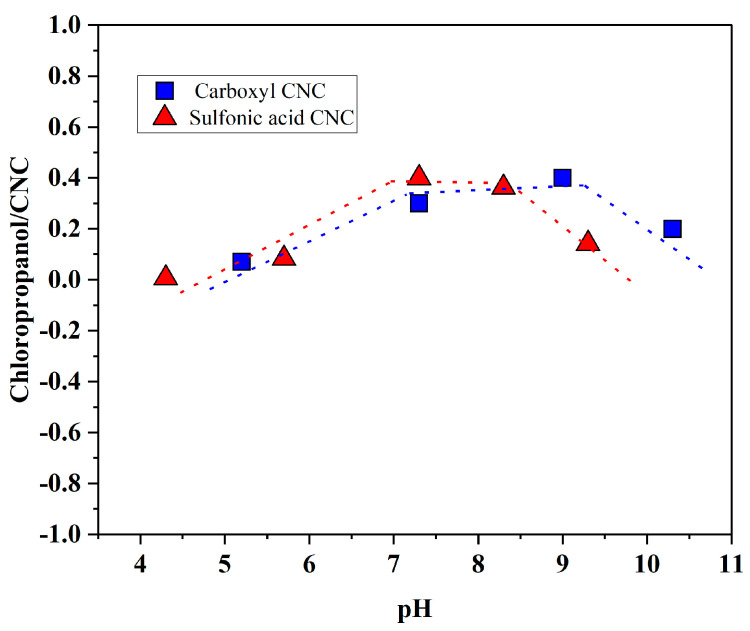
The effect of pH on the adsorption of chloropropanol on CNC.

**Figure 7 polymers-14-01746-f007:**
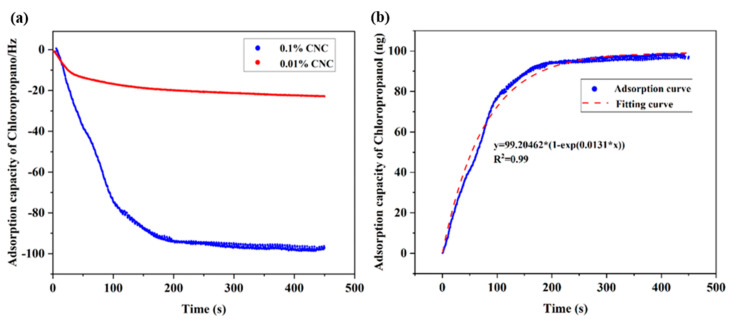
(**a**) Adsorption of chloropropanol at different CNC dosages. (**b**) Fitting curve of chloropropanol adsorption kinetics.

**Table 1 polymers-14-01746-t001:** Saturated adsorption capacity of chloropropanol on CNC.

Carboxylated CNC		Sulfonated CNC	
pH	Chloropropanol ng/CNC ng	pH	Chloropropanol ng/CNC ng
5.2	7 × 10^−2^	5.7	8.0 × 10^−2^
7.3	3 × 10^−1^	7.3	4.0 × 10^−1^
9.0	4 × 10^−1^	8.3	3.6 × 10^−1^
10.3	2 × 10^−1^	9.3	1.4 × 10^−1^

## Data Availability

Not applicable.

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
