# Peer review of "Adsorption Mechanism of Chloropropanol by Crystalline Nanocellulose"

_polymers, 2022, doi:10.3390/polym14091746_

Round 1

Reviewer 1 Report

The manuscript is related to nanocellulose which the authors did not extract. Although the special issue is related to cellulose and its derivatives, the paper does not specifically fit the scope. There are other issues related to significant flaws:

The first paragraph of the introduction is too general.

Line 26, what is “anti-corrosive”?

The second paragraph is too long, consider separating out some concepts.

There is no clear statement of the aims.

There is no indication of numbers of replicates and no error bars in the figures. The significance of the results is therefore not validated.

Author Response

Thank you for your comments. I have revised it as required

The manuscript is related to nanocellulose which the authors did not extract. Although the special issue is related to cellulose and its derivatives, the paper does not specifically fit the scope. 

The keywords in the journal special issue contain nanocellulose. In this paper, nanocellulose was used as the substrate material and the quartz microbalance technology was used to explore the adsorption mechanism and kinetics of chloropropanol by nanocellulose. I think it still fits the requirements of the special issue.

The first paragraph of the introduction is too general.

The first paragraph mainly introduces the advantages and disadvantages of plastic packaging, the prospect of paper packaging, and the advantages and disadvantages of plastic packaging are well known. Paper materials as packaging materials are also a new trend. The main research in this paper is the adsorption mechanism of chloropropanol by nanocellulose, so less space is used.

Line 26, what is “anti-corrosive”?

Preservation in line 26 means that the plastic packaging has anti-corrosion properties

The second paragraph is too long, consider separating out some concepts.

The second paragraph in the manuscript has been split. The introduction of chloropropanol and the introduction of carboxyl and sulfonic acid groups were split

There is no clear statement of the aims.

Chloropropanol is widely concerned in food safety, and it is a recognized carcinogen. With the wide application of paper packaging, the content of chloropropanol in paper packaging materials has also attracted attention, so it is of positive significance to explore the adsorption of chloropropanol in pulp. In paper production, sulfonic acid group and carboxyl group are common groups in pulp, so CNC of sulfonic acid group and carboxyl group is selected.

In this study, all experiments were repeated three times. Each point in Figure 5 in the manuscript represents a set of data from a quartz microbalance. The quartz microbalance records data every 0.3s, and each set of data has more than 10,000 data, so the experimental data is reliable and repeatable. Error analysis is also performed in 3.2 of the manuscript.

Reviewer 2 Report

Observations:

- The authors present an exhaustive characterization of crystalline nanocellulose (CNC) that was used as model to study the adsorption mechanism of chloropropanol. They found that physical adsorption was the prevailing process, the intermolecular hydrogen bonds acting as the driving force. The adsorption performance of sulfonated and carboxylated nanocellulose was also assessed, without presentation of these material, except diameter and length of fiber. More information about cellulose derivatives must be provided.

- The authors must evaluate the specific surface area, total, pore volume and average pore diameter of the samples in order to correlate these parameters with the adsorption phenomena.

- The figures quality must be improved.

- Please check the language.

Author Response

Thank you for your comments. I have revised accordingly.

 More information about cellulose derivatives must be provided.

The raw material of nanocellulose is cotton, the content is 8%, and the reagent grade.

- The authors must evaluate the specific surface area, total, pore volume and average pore diameter of the samples in order to correlate these parameters with the adsorption phenomena.

The nanocellulose in the manuscript first forms a dense film on the SiO2 sensor, which is discussed in this paper in the morphology characterization of CNC. “A large number of micropores are distributed on the nanocellulose coating, with a diameter of approximately 0.05 μm, as depicted in Figures 1b and 1c. This can be attributed to the large number of hydroxyl groups in CNC. The high density of hydroxyl groups allows CNC to be combined through hydrogen bonds. A strong porous structure is formed. Following the adsorption of CNC, the SiO2 sensor attains a certain capacity of physical adsorption. This is similar to the results obtained in previous research”

- The figures quality must be improved.

High-resolution images are reinserted

- Please check the language.

The language in the manuscript has been re-corrected.

Round 2

Reviewer 1 Report

Thank you for your replies and the edits. I still have some minor concerns. 

Please change anti-corrosive (line 26), this is not the correct term to describe plastics and is more descriptive of metals. A better and more general term would be "protective" or "offers protection".

The aims are still unclear, you did not attempt to edit this part of the introduction. This is necessary for better clarity for the readers.

In Table 1, you need a space between "Chloropropanolng" in both header columns.

Author Response

Thanks for your comments, I have made changes based on your comments.

Please change anti-corrosive (line 26), this is not the correct term to describe plastics and is more descriptive of metals. A better and more general term would be "protective" or "offers protection".

Modifications have been made in the manuscript. Line 26 is modified to “Plastic packaging offers protection and has a low packaging weight.”

The aims are still unclear, you did not attempt to edit this part of the introduction. This is necessary for better clarity for the readers.

Clarified the aim of the article and added“Therefore, this provides a reference for the application of paper packaging materials in food packaging materials. To explore the adsorption capacity and adsorption mechanism of chlorine-containing organic compounds in paper materials, and to provide a theoretical basis for reducing the adsorption capacity of chlorine-containing organic compounds in paper materials.”

In Table 1, you need a space between "Chloropropanol ng" in both header columns.

Modifications have been made in the manuscript.

Reviewer 2 Report

The evaluation of specific surface area, pore volume and average pore diameter of the samples could improve the discussion part, as well as conclusion of this paper.

Author Response

Thanks for your comments, I have made changes based on your comments.